# Multilayer Dielectric Periodic Antenna Structure in a Cascade View

## Marian Wnuk

Faculty Electronics, Military University of Technology, 00-908 Warsaw, Poland; marian.wnuk@wat.edu.pl;
Tel.: +48-601-819-691

**Abstract:** The spectral response of the periodic antenna structure placed in a dielectric homogeneous medium depends on the antenna geometry, the parameters of the medium, the angle of incidence, polarization, and the geometry of the excitation field. Increasing the number of antenna structure parameters can be achieved by introducing a multilayer dielectric medium with a certain number of metallized periodic surfaces located on flat boundaries between the dielectric layers. There are two complementary approaches to the analysis of such structures. In the first, the composite antenna system is analysed by constructing supermodes of the entire structure. In the second, the system is considered as a cascade assembly of flat discrete elements, i.e., the boundaries between two dielectrics, periodic metallized planes, and dielectric layers. The latter approach leads to the definition of the scattering, transmission, or impedance matrix of the entire structure by cascading the corresponding matrices associated with the individual discrete elements of the antenna structure. It is particularly useful in modelling dielectric multilayer antenna walls, where the stored data on one planar antenna element can be used many times in the analysis of various antenna systems with modified parameters of other discrete structure elements. Microstrip antennas combine field and peripheral problems and require the use of analytical methods of a high degree of complexity. Therefore, at present, there are no standard methods that can be used in engineering practice. This work is a step towards filling these gaps.

**Keywords:** harmonics Floquet; periodic antenna multilayer dielectric

## 1. Introduction

The spectral response of the periodic antenna structure placed in a dielectric homogeneous medium depends on the antenna geometry, the parameters of the medium, the angle of incidence, polarization, and the geometry of the excitation field. However, in many applications, the above number of parameters determining the operation of the antenna is not sufficient to be able to model its desired directional characteristics. Increasing the number of antenna structure parameters can be achieved by introducing a multilayer dielectric medium with a certain number of metallized periodic surfaces located on flat boundaries between the dielectric layers [1,2]. The analysis of such a multilayer planar antenna system is the subject of this work. There are two complementary approaches to the analysis of such structures. In the first, the composite antenna system is analysed by constructing supermodes of the entire structure. In the second, the system is considered as a cascade assembly of flat discrete elements, i.e., the boundaries between two dielectrics, periodic metallized planes, and dielectric layers [3]. The latter approach leads to the definition of the scattering, transmission, or impedance matrix of the entire structure by cascading the corresponding matrices associated with the individual discrete elements of the antenna structure. It is particularly useful in modelling dielectric multilayer antenna walls, where the stored data on one planar antenna element can be used many times in the analysis of various antenna systems with modified parameters of other discrete structure elements.

In practice, antenna arrays most often consist of periodic metal structures placed on or immersed in a dielectric multilayer medium. Changes in the size and shape of individual

antenna elements enable efficient modeling of its spectral parameters. These changes can be additionally modeled by selecting the geometrical and physical parameters of the dielectric layered substrate and coating. In the case of large transverse dimensions, the antenna is in the order of 100 wavelengths and several dozen or more individual metal radiation elements. The radiation properties of the antenna approximately correspond to those of the unlimited antenna structure. Under the condition of periodicity of such an antenna and its excitation with linear, homogeneous phase modulation, the problem of scattering on such a structure comes down to the analysis of one basic antenna cell, usually in a spectral space spanned by a complete array of corresponding Floquet harmonics. For structures of smaller sizes and fewer elements, the effects related to their limitations, especially the edge effects occurring on their outer cells, may affect the antenna performance characteristics and should be considered in their modeling. So far, there is no known exact effective method for solving the scattering problem on such periodic constrained structures. Attempts to solve this problem by directly applying the spectral method of moments are only formally strict. This is because their accuracy is limited by the constraints imposed on the size of the matrices used in the procedures for solving them. Nevertheless, for antenna walls composed of a few or a dozen regular-shaped basic cells, this method gives good results. Other known methods of solving the scattering problem on larger periodic structures, such as modifications of the above-mentioned direct method by limiting the number of base functions and appropriate selection of their waveforms, or the iterative method imposed on the solution by successive application of the Fast Fourier transform, are approximate, slow convergent, and can only be used in some simple, easy-to-analyze cases. Currently, a computer program is being developed that will allow verification of the adopted assumptions.

The generalized scattering matrix method was originally used in the analysis of waveguide problems, considering both propagating and vanishing [4]. Treating the composite antenna array as a waveguide with many planar discontinuities, a similar analysis can also be applied to antenna theory. This method is applied in this work to multilayer periodic systems in one or two directions.

## 2. Statement of the Problem

We consider the composite periodic antenna structure shown in Figure 1. The system consists of many-layered elements bounded at the top and bottom by half-spaces (upper and lower) of homogeneous dielectric media. The third axis ($y$) of the orthogonal coordinate system ($x,y,z$) is perpendicular to the drawing plane. The basic components of the structure are: metallized, infinitely periodic in the plane ($x,y$) antenna surface, the boundary between two dielectrics, and a homogeneous dielectric layer [5,6].

The periodic flat antenna structure is shown in Figure 1. The geometry of a single antenna element remains arbitrary. The distance between the individual antenna elements in the $x$ and $y$ directions is $d_x$ and $d_y$, respectively. The inclination angles of the symmetry axis of the antenna structure with respect to the $x$ axis are respectively 0 and $\Omega = 90° - \Theta$. Let us assume that the designations of the multi-layer antenna system are shown in Figure 2.

The matrix $\underline{S}^{(j)}$ or $\underline{T}^{(j)}$ represents the planar discrete element of the antenna in the form of a flat metallized grid, a boundary between two dielectric media or a dielectric layer [7]. It was assumed that $a'_j = b_{j+1}$, $a_j = b'_{j+1}$.

Vectors, $a_j = \left[ a_{j1}, a_{j2}, \ldots a_{jJ_j} \right]$ $b_j = \left[ b_{j1}, b_{j2}, b_{jJ_j} \right]$ $a'_j = \left[ a'_{j1}, a'_{j1}, \ldots a'_{jJ_j} \right]$, $b'_j = \left[ b'_{j1}, b'_{j1}, b'_{jJ_j} \right]$ are the complex amplitudes of the normalized incoming $\left( a_j, a'_j \right)$ and outgoing $\left( b_j, b'_j \right)$ discrete modes of a single structure, represented by a scattering matrix $\underline{S}^{(j)}$, or a transmission matrix $\underline{T}^{(j)}$. The matrix $\underline{S}^{(j)}$, binds the outgoing $\left( b_j, b'_j \right)$ modes with falling modes $\left( a_j, a'_j \right)$

$$\begin{bmatrix} b_j \\ b'_j \end{bmatrix} = \begin{bmatrix} \underline{\underline{S}}^{(j)} \end{bmatrix} \begin{bmatrix} a_j \\ a'_j \end{bmatrix} = \begin{bmatrix} \underline{S}^{(j)}_{11} & \underline{S}^{(j)}_{12} \\ \underline{S}^{(j)}_{21} & \underline{S}^{(j)}_{22} \end{bmatrix} \begin{bmatrix} a_j \\ a'_j \end{bmatrix} \tag{1}$$

and matrix $\underline{T}$ connects the field on the "right $a_j^{(c)}$, $b_j^{(c)}$, $a_N^{(c)}$, $b_N^{(c)}$", lower side of the structure with the field on the "left", upper side of the structure:

$$
\begin{bmatrix} a_j^{(c)} \\ b_j^{(c)} \end{bmatrix} = \left[ \underline{T}^{(j)} \right] \begin{bmatrix} a_j \\ a_j^{(c)} \end{bmatrix} = \begin{bmatrix} \underline{T}_{11}^{(j)} & \underline{T}_{12}^{(j)} \\ \underline{T}_{21}^{(j)} & \underline{T}_{22}^{(j)} \end{bmatrix} \begin{bmatrix} a_j \\ b_j \end{bmatrix}
\tag{2}
$$

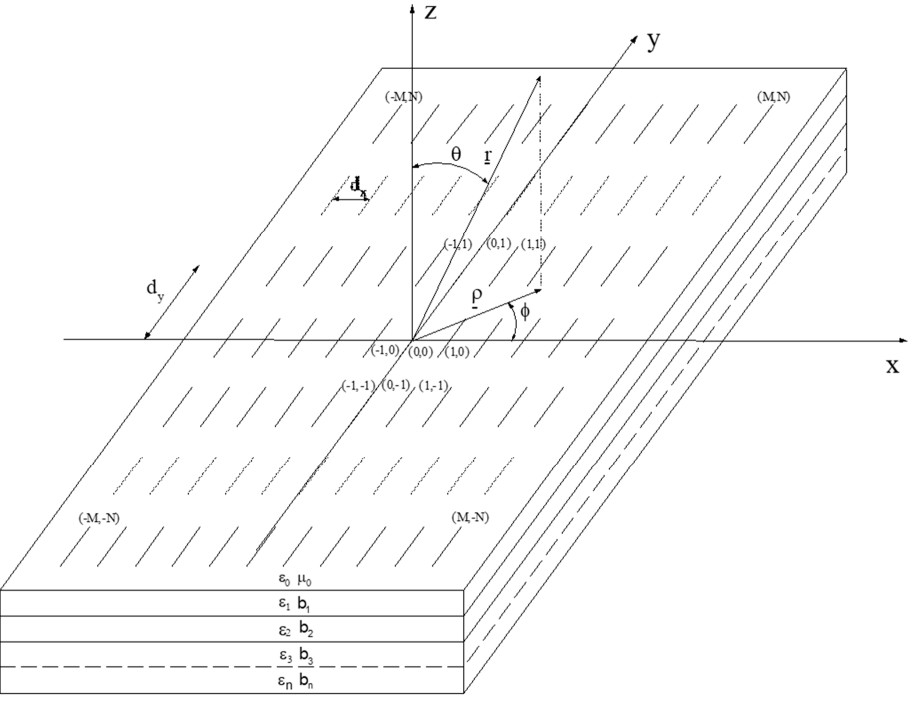

**Figure 1.** Flat periodic antenna structure infinite in the (*x,y*) plane, multilayer structure.

**Figure 2.** Block diagram of a multi-layer antenna.

Both types of matrices lead to a global description of the operation of the entire antenna structure [8]. Global transmission matrix $\underline{T}$

$$
\underline{T} \begin{bmatrix} a_1 \\ b_1 \end{bmatrix} = \begin{bmatrix} a_N^{(c)} \\ b_N^{(c)} \end{bmatrix}
\tag{3}
$$

has the form of a cascade assembly of individual elements of the antenna structure, which corresponds to the product of the transmission matrix of individual structure elements:

$$
\underline{\underline{T}} = \underline{\underline{T}}^{(J)} \underline{\underline{T}}^{(J-1)} \dots \underline{\underline{T}}^{(2)} \underline{\underline{T}}^{(1)}
\tag{4}
$$

Knowing the matrices $T^{(j)}$ and $T$, it is possible to determine the matrices $S^{(j)}$ and $S$ according to the formula:

$$[\underline{S}] = \begin{bmatrix} \underline{S}_{11}^{(j)} & \underline{S}_{12}^{(j)} \\ \underline{S}_{21}^{(j)} & \underline{S}_{22}^{(j)} \end{bmatrix} = \begin{bmatrix} -\underline{T}_{22}^{-1}\underline{T}_{21} & \underline{T}_{22}^{-1} \\ \underline{T}_{11} - \underline{T}_{12}\underline{T}_{22}^{-1}\underline{T}_{21} & \underline{T}_{12}\underline{T}_{22}^{-1} \end{bmatrix} \qquad (5)$$

Then we determine the global transmission matrix by the product of successive partial transmission matrices. Thus, for the determination of the scattering matrices it is practically necessary also to determine the partial and global transmission matrices. Similarly, knowing the form of the scattering matrices $\underline{S}^{(j)}$ of individual elements of the structure, the matrices $T^{(j)}$ can be determined according to the formula:

$$\left[\underline{T}^{(j)}\right] = \begin{bmatrix} \underline{T}_{11}^{(j)} & \underline{T}_{12}^{(j)} \\ \underline{T}_{21}^{(j)} & \underline{T}_{22}^{(j)} \end{bmatrix} = \begin{bmatrix} \underline{S}_{21}^{(j)} - \underline{S}_{22}^{(j)}\underline{S}_{12}^{(j)-1}\underline{S}_{11}^{(j)} & \underline{S}_{22}^{(j)}\underline{S}_{12}^{(j)-1} \\ -\underline{S}_{21}^{(j)-1}\underline{S}_{11}^{(j)} & \underline{S}_{12}^{(j)-1} \end{bmatrix} \qquad (6)$$

and hence we also obtain the global transmission matrix $\underline{T}$ as a product of the matrix $T^{(j)}$ [7].

In general, the cut-off numbers $K_j$ of an infinite countable number of modes of a structure may change from one subsystem of the structure to another, i.e., they may depend on the number $j$. For the simplicity of notation and further numerical implementation, however, assume that $K_j = const.$ for any subsystem $j = 1, 2, \ldots J$, which is also commonly used in numerical calculations. So we assume that the matrices $\underline{S}^{(j)}$ and $\underline{T}^{(j)}$ are square of the order $2K$ and the matrices $\underline{S}^{(j)}{}_{kl}$, $\underline{T}^{(j)}{}_{kl}$, $k, l = 1, 2$ are of the order $K$ [9]. Particular antenna subsystems, as well as the entire antenna system, can also be expressed by impedance matrices $\underline{Z}^{(j)}$ or admittance matrices $\underline{Y}^{(j)}$ according to the following formulas [4,5] ($[S] \equiv \underline{S}$):

$$[Z] = \frac{[1] + [S]}{[1] - [S]}[\varsigma_0] = \left[Y^{-1}\right] \qquad (7)$$

$$[S] = \frac{[Z]\left[\varsigma_0^{-1}\right] - [1]}{[Z]\left[\varsigma_0^{-1}\right] + [1]} = \frac{[1] - [\varsigma_0][Y]}{[1] + [\varsigma_0][Y]} \qquad (8)$$

where: $[\varsigma_0]$ diagonal matrix of characteristic impedances.

The form of the matrix $\underline{S}$, $\underline{T}$, $\underline{Z}$, or $\underline{Y}$ depends on the geometry and parameters of the antenna structure medium, as well as on the adopted complete base of eigenvectors of the appropriate boundary problem. The electromagnetic field at any point in the structure is presented in the form of Fourier integrals or series with appropriate, generally complex, coefficients (amplitudes) of the distribution [10,11]. For infinite periodic structures, the equivalent of Fourier transforms is the field distributions to Floquet's harmonics. As a result, the transverse electric field $\underline{E}_t$ is presented in the form of a series:

$$\begin{aligned} \underline{E}_t(x,y,z) &= \sum_k v_k^{\pm}\underline{\Psi}_k exp(\pm ik_{zk}z) = \sum_k v_k^{\pm}\underline{\Psi}_k e_k(x,y)exp(\pm ik_{zk}z) \\ &= \sum_k v_k^{\pm}\underline{\Psi}_k\left[ik_{xk}x + ik_{yk}y\right]v_k^{\pm}\underline{\Psi}_k\left[ik_{xk}x + ik_{yk}y\right] \end{aligned} \qquad (9)$$

where $v_k^{\pm}$ denote the coefficients of the field distribution propagating in a positive (+) or negative (−) direction with a phase factor $exp(\pm ik_{zk}z)$. The vectorial normalized to the unit power of Floquet's harmonics $\Psi_k$, defined in the next subsection, have a phase factor $e_k(x,y)$ and vector amplitudes $\psi_k$. The index $k$ follows TE or TM polarization and the generally infinite Floquet harmonics indices in the $x(m)$ and $y(n)$ directions [12]. Similarly, for the transverse magnetic field we get:

$$\underline{H}_t(x,y,z) = \sum_k v_k^{\pm}\left(\pm\eta_k^{-1}\right)(\underline{n}_z \times \underline{\Psi}_k)exp(\pm ik_{zk}z) \qquad (10)$$

where: $\eta_q$ denotes the harmonic modal impedance with index $q$. With the normalization of the power of individual harmonics adopted below, the total field power flux in the $z$ direction is expressed by the amplitude modulus $v_k$:

$$P_z(z) = \sum_k |v_k^\pm|^2 exp(\mp 2Im(k_{zk})z) \tag{11}$$

where the summation along the index $k$ runs both the propagating modes ($Im(k_{zk}) = 0$) and vanishing modes ($Im(k_z k) \neq 0$) [13].

### 3. Modes of the Planar Endless Bidinary Periodic Structure Scalar Harmonics of Floquet

Let us consider the function $f(x,y)$ periodically depending on $x$ and $y$ with the accuracy to the linear phase factor $\varphi_x + \varphi_y$:

$$f(x + d_x, y + d_y) = f(x,y)exp[-i(\varphi_x + \varphi_y)] \tag{12}$$

where $d_x$ and $d_y$ are lattice constants in the plane $(x,y)$ and the function $f(x,y)$ satisfies the Helmholtz equation in space $(x,y)$ [2]:

$$\left(\nabla_t^2 + k_x^2 + k_y^2\right)f(x,y) = 0 \tag{13}$$

Moreover, the longitudinal components $kz$ and the transverse components $k_x$ and $k_y$ of the propagation vector satisfy the dispersion equation: $k_x^2 + k_y^2 + k_z^2 = k^2$.

The above definition of the function $f(x,y)$ shows that the function $F(x,y) = f(x,y) exp[i(\varphi_x x/d_x + \varphi_y y/d_y)]$ is periodic, i.e., it satisfies the condition $F(x + d_x, y + d_y) = F(x,y)$ and as such can be represented as a typical Fourier series:

$$F(x,y) = \sum_{m,n} A_{mn} exp[i2\pi(mx/d_x + ny/d_y)]. \tag{14}$$

Hence the pseudo-periodic function $f(x,y)$ is represented in the form of a distribution [14]

$$\begin{aligned} f(x,y) &= F(x,y)exp[-i(\varphi_k x/d_x - \varphi_y y/d_y)] \\ &= \sum_{m,n} A_{m,n} exp[i(k_{xm}x + k_{ym}y)] = \sum_{m,n} A_{mn} e_{mn}(x,y) \end{aligned} \tag{15}$$

for Floquet harmonics:

$$e_{mn}(x,y) = exp[i(k_{xm}x + k_{yn}y)], \tag{16}$$

with the coefficients of the $A_{mn}$. distribution. The form of the components of the propagation vector $\underline{k}$ reflects the periodic symmetry of the problem under consideration along the $x$ and $z$ axes:

$$k_{xm} = 2\pi m/d_x - \varphi_x/d_x, \tag{17}$$

$$k_{yn} = 2\pi n/d_y - \varphi_y/d_y, \tag{18}$$

$$k_{zmn} = \left[k^2 - k_{xm}^2 - k_{ym}^2\right]^{1/2}. \tag{19}$$

Note that the transverse component (in the $x,y$ plane) of the wave vector is:

$$k_{rmn} = \left[k^2 - k_{zmn}^2\right]^{1/2} = [k_{xmn}^2 + k_{ymn}^2]^{1/2}. \tag{20}$$

The above distribution is a generalization of the Fourier series of the pseudo-periodic function $f(x,y)$ with linear phase increment $(m\varphi_x + n\varphi_y)$ for translation $(mx + ny)$ in the transverse plane $(x,y)$ [15] For a two-dimensional periodic structure with the inclination

angle $\Omega$ of the second symmetry axis with respect to the $x$ axis (the remaining symmetry axis coincides with the $x$ axis), we get:

$$k_{xmn} = \frac{2\pi n}{d_x} - \frac{2\pi m}{d_y} - \varphi_y / d_y. \tag{21}$$

$$k_{ymn} = \frac{2\pi n}{d_y sin\Omega} - \frac{2\pi m}{d_y tan\Omega} - \varphi_y / d_y. \tag{22}$$

In the general case, for any position of the axis of symmetry with respect to the axis of orthogonal coordinates $x$ and $y$, all three components of the propagation vector $k_{xmn}$, $k_{ymn}$, $k_{zmn}$ depend on both $m$ and $n$. We then make these components dependent on the wave vector of the zero ($m = n = 0$) harmonics Floquet of the incident field

$$k_x = k_x^{inc} = ksin\Theta^{inc} \cdot cos\varphi^{inc} \tag{23}$$

$$k_y = k_y^{inc} = ksin\Theta^{inc} \cdot sin\varphi^{inc} \tag{24}$$

where $\theta^{inc}$ and $\varphi^{inc}$. denote the angles of incidence in the planes $(z,y)$ and $(x,y)$, respectively. Then the phase shifts $\varphi_x$, $\varphi_y$ for the translation of $d_x$, $d_y$ by one periodic cell are respectively:

$$\varphi_x = -k_x^{inc} \cdot d_x \tag{25}$$

$$\varphi_y = -k_y^{inc} \cdot d_y \tag{26}$$

$$k_{xmn} = \frac{2\pi m}{d_x} + k\,sin\Theta^{inc} \cdot cos\varphi^{inc} \tag{27}$$

$$k_{ymn} = \frac{2\pi n}{d_y sin\Omega} - \frac{2\pi m}{d_y tan\Omega} + k\,sin\Theta^{inc} \cdot sin\varphi^{inc} \tag{28}$$

Without losing the generality of considerations, let us then assume that all individual components ($j$) of the layered antenna structure are defined with respect to the upper outer center, i.e., the incidence center. Then, for any $j$, from the continuity of the tangent components $E_t = (E_x,E_y)$, $H_t = (H_x,H_y)$ in the planes of discontinuity $z = z_j = const.$ it follows that the transverse and correspond to the transverse components of the wave vector in the following form:

$k_{xmn} = \frac{2\pi m}{d_x} + k\,sin\Theta^{inc} \cdot cos\varphi^{inc} k_{x00}$, $k_{y00}$ components of the Floquet harmonic wave vector of the zero spectral cell ($m = 0$, $n = 0$) remain equal to the corresponding component in the incidence center:

$$k_{x00} = k\,sin\Theta^{inc} \cdot cos\varphi^{inc}, \tag{29}$$

$$k_{y00} = k\,sin\Theta^{inc} \cdot sin\varphi^{inc}. \tag{30}$$

The above property holds true for any $j$, i.e., for each individual element of the composite antenna structure. In other words, the components $k_{x00}$, $k_{y00}$ remain independent of $z$. Similarly, for any values of $m$ and $n$, i.e., for harmonics of any order, we get:

$$k_{xmn} = \frac{2\pi m}{d_x} + k_{x00}, \tag{31}$$

$$k_{ymn} = \frac{2\pi n}{d_y\,sin\Omega} - \frac{2\pi m}{d_y tan\Omega} + k_{y00}. \tag{32}$$

Let us then proceed to deriving the representation for the transverse components of the electromagnetic field in the form of the distribution into Floquet vector harmonics.

## 4. Vector Harmonics of Floquet—Electromagnetic Field Representation in the Plane (*x,y*)

The electromagnetic field at any point of the previously defined multi-layer antenna structure can be decomposed into two independent field components distinguished by their polarization with respect to the antenna plane normal, i.e., TE field with an electric field vector perpendicular to the *z* direction: $\underline{E}(x,y,z) = (\underline{E}_t, E_z)$ and the TM field with the magnetic field vector perpendicular to the *z* direction: $\underline{H}(x,y,z) = (\underline{H}_t, H_z)$. As part of this division, we will define the distributions of the transverse electric field into vector Floquet harmonics, characterizing the mode structure of the electromagnetic field of an unlimited, multi-layer periodic structure. It should be noted that with the assumed losslessness of the upper (incident) medium, the above-analyzed tangents (to the (*x,y*) plane) coordinates of the propagation vector corresponding to the incident field harmonics remain real in every element of the antenna structure, both lossy and lossless.

### 4.1. TE Polarization

For the TE field, let us assume that the longitudinal component of the magnetic field $H_{zTEmn}$ as a pseudo-periodic function $f(x,y)$ with the appropriate phase factor depending on *z*:

$$H_{zTEmn}(x,y,z) = f(x,y)exp[ik_{zTEmn}z]. \tag{33}$$

From Maxwell's equations, the electric transverse field $E_{tTEmn} = (E_{xTEmn}, E_{yTEmn})$ is expressed through the field $H_{zTEmn}$ [16] as follows:

$$\underline{E}_{tTEmn} = \frac{i\omega\mu}{k_{yTEmn}^2}\underline{n}_z \times \nabla_t H_{zTEmn} = \frac{i\omega\mu}{k_{xTEmn}^2 + k_{yTEmn}^2}\underline{n}_z \times i\left(\underline{n}_x k_{xTEmn} + \underline{n}_y k_{yTEmn}\right)H_{zTEmn} \tag{34}$$

where: $\underline{n}_x$, $\underline{n}_y$, $\underline{n}_z$ denote the unit vectors along the respective *x*, *y*, and *z* axes. In further considerations, we will also denote the sequence of *TEmn* or *TMmn* pointers with one symbol from the set {*q,p*}. Let us introduce the orthogonal and normalized Floquet vector harmonics $\Psi_q$ ($q = \Psi_q$ ($q = TEmn$)), i.e., satisfying the condition:

$$\pm \underline{n}_z \circ \iint_S \underline{\psi}_{TEm'n'} \times \underline{\psi}_{TEmn}^* \, dxdy = \delta_{mm'}\delta_{nn'} \tag{35}$$

where integration takes place over a single elementary cell of a periodic structure with the surface of *S*. The normalized Floquet harmonics contain, in addition to the phase factor

$$e_{TEmn}(x,y) = exp[i(k_{xTEmn}x + k_{yTEmn}y)] \tag{36}$$

also normalizing vector amplitude coefficient:

$$\underline{\psi}_q(k_{xq},k_{yq}) = \underline{n}_x\psi_{xTEmn} + \underline{n}_y\psi_{xTEmn} = (d_xd_y)^{-1/2}\left[\frac{\underline{n}_x k_{yq} - \underline{n}_y k_{xq}}{(k_{xq}^2 + k_{yq}^2)^{1/2}}\right] . \tag{37}$$

Thus, Floquet's vector harmonics take the following form:

$$\underline{\Psi}_q = \underline{\psi}_q(k_{xq},k_{yq})e_q(x,y) = (d_xd_y)^{-1/2}\left[\frac{\underline{n}_x k_{yq} - \underline{n}_y k_q}{(k_{xq}^2 + k_{yq}^2)^{1/2}}\right]exp[i(k_{xq}x + k_{yq}y)] \tag{38}$$

with coordinates in the *x* and *y* directions, respectively:

$$\Psi_{xq}(k_{xq},k_{yq}) = +(d_xd_y)^{-1/2}\left(k_{xq}^2 + k_{yq}^2\right)^{-1/2}k_{yq}exp[i(k_{xq}x + k_{yq}y)] \tag{39}$$

$$\Psi_{yq}(k_{xq},k_{yq}) = -(d_xd_y)^{-1/2}\left(k_{xq}^2 + k_{yq}^2\right)^{-1/2}k_{xq}exp[i(k_{xq}x + k_{yq}y)] \tag{40}$$

Finally, the transverse electric field with a polarization *TE* $\underline{E}_{tq}$ ($q = TEmn$) is defined as proportional to the transverse vector harmonic $\Psi_{tq}$:

$$\underline{E}_{tq}(x,y) = (\eta_q^*)^{+1/2}\underline{\Psi}_q(x,y) \tag{41}$$

and the longitudinal component $\underline{H}_{tq}$ of the magnetic field with a polarization *TE* is determined from the knowledge of the transverse electric field $E_{tq}$

$$\underline{n}_z \times \underline{H}_{tq} = \pm\eta_q^{-1}\underline{E}_{tq} \tag{42}$$

$$\underline{n}_z \times \underline{E}_{tq} = \pm\eta_q\underline{H}_{tq} \tag{43}$$

and using the mode impedance $\eta_q$ of a harmonic with a polarization *TE*.

The factor $(\eta_q^*)^{1/2}$ standing at the normalized vector harmonic $\Psi_q$ provides the normalization of the power of the harmonic field, which corresponds to the unit power flow conducted through this harmonic in the direction of the *z* axis:

$$\pm\,\underline{n}_z \circ \iint_S \underline{E}_{tq} \times \underline{H}_{tq'}^* \, dxdy = \underline{n}_z \circ (\eta_q^*)^{-1} \iint_S \underline{E}_{tq} \times \left(\underline{n}_z \times \underline{E}_{tq'}^*\right) dxdy = \delta_{qq'} \tag{44}$$

*4.2. TM Polarization*

For the TM field, let us take the longitudinal component of the electric field $E_{zTMmn}$ as the functions $f(x,y)$ with the appropriate phase factor depending on *z*:

$$E_{zTMmn}(x,y,z) = f(x,y)exp[ik_{zTMmn}z] \tag{45}$$

Transverse fields: magnetic $\underline{H}_{tTMmn} = (H_{xTMmn}, H_{yTMmn})$ and electric $\underline{E}_{tTMmn} = (E_{xTMmn}, E_{yTMmn})$ are expressed through the $E_{zTMmn}$ field as follows [6,17]:

$$\begin{aligned}\underline{H}_{tTEmn} &= \frac{i\omega\varepsilon}{k_{xTMmn}^2+k_{yTMmn}^2}\underline{n}_z \times \nabla_t H_{zTMmn} \\ &= \frac{i\omega\varepsilon}{k_{xTMmn}^2+k_{yTMmn}^2}\underline{n}_z \times i\left(\underline{n}_x k_{xTMmn} + \underline{n}_y k_{yTMmn}\right)H_{zTMmn}\,,\end{aligned} \tag{46}$$

$$\underline{E}_{tTMmn} = \frac{-ik_{zTMmn}}{k_{xTMmn}^2 + k_{yTMmn}^2}\nabla_t E_{zTMmn}. \tag{47}$$

Let us introduce the orthonormal vector harmonics of Floquet [18]: $\Psi_{TMmn} \equiv \Psi_p$ ($p = TMmn$) about TM polarization

$$\underline{\Psi}_p = (d_x d_y)^{-1/2}\left[\frac{\underline{x}k_{xp} + \underline{y}k_{yp}}{(k_{xp}^2 + k_{yp}^2)^{1/2}}\right]exp[i\left(k_{xp}x + k_{yp}y\right)], \tag{48}$$

that is

$$\psi_{xTMmn}\left(k_{xk}, k_{yk}\right) = +\left(d_x d_y\right)^{-1/2}\left(k_{xk}^2 + k_{yk}^2\right)^{-1/2}k_{xTMmn}, \tag{49}$$

$$\psi_{yTMmn}\left(k_{xk}, k_{yk}\right) = +\left(d_x d_y\right)^{-1/2}\left(k_{xk}^2 + k_{yk}^2\right)^{-1/2}k_{yTMmn}. \tag{50}$$

Then, analogically to the case of *TE* polarization, the orthogonality relationship takes place:

$$\pm\,\underline{n}_z \circ \iint_S \underline{\Psi}_{Mm'n'} \times \underline{\Psi}_{TMmn}^* \, dxdy = \delta_{mm'}\delta_{nn'} \tag{51}$$

Additionally, *TE* modes remain orthogonal to *TM* modes:

$$\iint_S \underline{\Psi}_{Mm'n'} \times \underline{\Psi}_{TEmn}^* \, dxdy = 0 \tag{52}$$

which leads to the relationship of orthogonality in the general form ($p$, $q$ run for any values from the sets $\{TE, TM\}$, $\{m\}$, $\{n\}$):

$$\pm \underline{n}_z \circ \iint_S \underline{\Psi}_p \times \underline{\Psi}_q^* \, dxdy = \delta_{pq}. \tag{53}$$

Then, considering the mode impedance for the polarization we finally obtain the form of the transverse components [19] of the electric field $\underline{E}_{tp}$ and magnetic $\underline{H}_{tp}$ with *TM* polarity ($p = TMmn$):

$$\underline{E}_{tp}(x,y) = (\eta_p)^{+1/2} \underline{\Psi}_p(x,y) \tag{54}$$

$$\pm \eta_p \underline{H}_{tp} = \underline{n}_z \times \underline{E}_{tp} \tag{55}$$

$$\pm \eta_p^{-1} \underline{E}_{tp} = \underline{n}_z \times \underline{H}_{tp}. \tag{56}$$

Hence the transverse magnetic field $\underline{H}_{tp}$ is expressed by vector harmonics $\underline{H}_{tp}$ in the following way:

$$\underline{H}_{tp}(x,y) = \pm \eta_p^{-1} \underline{n}_z \times \underline{E}_{tp}(x,y) = \pm \eta_p^{-1/2} \underline{n}_z \times \underline{\Psi}_p(x,y) \tag{57}$$

The coefficient $(\eta_p)^{-1/2}$ at the normalized vector harmonic $\underline{n}_z \times \underline{\Psi}_p$ ensures the normalization of the harmonic field power with the polarization *TM*:

$$\pm \underline{n}_z \iint_S \underline{E}_{tp} \times \underline{H}_{tp'}^* \, dxdy = \underline{n}_z (\eta_p)^{+1} \iint_S \left( \underline{n}_z \times \underline{H}_{tp} \right) \times \underline{H}_{tp'}^* \right) dxdy = \underline{n}_z \delta_{pp'} \tag{58}$$

## 5. General Dispersion Material for Individual Composite Antenna Structure

Based on the mode structure defined in the previous section, the electromagnetic field of the composite antenna can, for any of the following, be presented in the form of a series:

$$\underline{E}_t(x,y,z) = \sum_k \left\{ v_k^+ \underline{\Psi}_k exp(+ik_{zk}z) + v_k^- \underline{\Psi}_k exp(-ik_{zk}z) \right\} \tag{59}$$

$$\underline{H}_t(x,y,z) = \underline{n}_z \times \left[ \sum_k \eta_k^{-1} \left\{ +v_k^+ \underline{\Psi}_k exp(+ik_{zk}z) - v_k^- \underline{\Psi}_k exp(-ik_{zk}z) \right\} \right] \tag{60}$$

with the known vector Floquet harmonics $\underline{\Psi}_k$ and the distribution coefficients remaining to be determined $v_k$ [20,21]. Given the known field incident on the antenna, determining the total field requires deriving the scattering matrix of the entire system. The solution is obtained by determining the scattering matrix of individual planar system elements, such as the dielectric boundary, metallized periodic plane, and a layer of a homogeneous dielectric medium. In each of these cases, the distribution coefficients $v_k$ correspond to the field amplitudes defined in Figure 2.

Let us consider any single element of a composite antenna denoted by the index $j$. Let us create a vector $\underline{V}_j^{(in)}$ of field amplitudes entering (*in*) into the system (*j*), which in turn consists of a vector $\underline{V}_j^{(inup)}$ of amplitudes of type in on the upper (upper) side of the system (*j*) and the vector $\underline{V}_j^{(inlow)}$ of amplitudes of type in from the lower side of the system (*j*):

$$\underline{V}_j^{(in)} = \left[ \begin{array}{c} \underline{V}_j^{(inup)} \\ \underline{V}_j^{(inlow)} \end{array} \right] = \left[ \begin{array}{c} \underline{a}_j \\ \underline{a}_j' \end{array} \right] . \tag{61}$$

Similarly, we define the vector $V_j^{(ou)}$ of field amplitudes coming from the system (*j*):

$$\underline{V}_j^{(ou)} = \left[ \begin{array}{c} \underline{V}_j^{(ouup)} \\ \underline{V}_j^{(oulow)} \end{array} \right] = \left[ \begin{array}{c} \underline{b}_j \\ \underline{b}_j' \end{array} \right] . \tag{62}$$

The definition of the relationship between these two vectors is determined by the scattering matrix $\underline{S}^{(j)}$:

$$\left[ \underline{V}_j^{(ou)} \right] = \left[ \begin{array}{cc} \underline{S}_{11}^{(j)} & \underline{S}_{12}^{(j)} \\ \underline{S}_{21}^{(j)} & \underline{S}_{22}^{(j)} \end{array} \right] \left[ \underline{V}_j^{(in)} \right] \tag{63}$$

So the vector $V_j^{(in)}$ consists of an infinite sequence of amplitudes from the top of the element (*j*), $v_k$—from the bottom of the element (*j*), and the vector $V_j^{(ou)}$—from the sequences of the respective amplitudes and

$$\underline{V}_j^{(in)} = \left[ \begin{array}{c} v_1^- \\ v_2^- \\ \vdots \\ v_1^+ \\ v_2^+ \\ \vdots \end{array} \right] \begin{array}{l} dla\ z \leq z_j \\ \\ \\ \\ dla\ z \geq z_j \end{array} \tag{64}$$

$$\underline{V}_j^{(ou)} = \left[ \begin{array}{c} v_1^+ \\ v_2^+ \\ \vdots \\ v_1^- \\ v_2^- \\ \vdots \end{array} \right] \begin{array}{l} dla\ z \leq z_j \\ \\ \\ \\ dla\ z \geq z_j \end{array} \tag{65}$$

Inequalities mean, conventionally, the top and bottom side of a single antenna element. For example, for the dielectric boundary and the periodic plane, this means $lim\delta \to 0\ (zj \pm \delta)$, respectively. In numerical calculations, the vectors $V_j^{(in)}$, $V_j^{(ou)}$ with an infinite number of parameters are usually cut to 2 (*M* + *N*) elements, where the number 2 corresponds to the two polarizations *TE* and *TM*, *M*—the number of harmonics in the *x* direction, and *N*—the number of harmonics in the *y* direction. It follows that *S* (*j*) *kl*, *k*, *l* = 1.2 are of the order of 2MN. In order to determine the global matrix *S*, it is necessary to know the scattering matrix of individual system elements. These include the metallized periodic plane, the dielectric boundary, and the homogeneous dielectric layer.

## 6. Plane with Endlessly Metal Periodic Mesh

The geometry of the periodic plane is presented in Figure 1. The geometry of a single periodic cell is arbitrary. As a result of the scattering analysis on such a structure (using the spectral variant of the Galerkin method) [22], we obtain an equation linking the vector of the scattered field amplitudes $V_{js}$ with the field amplitude vector incident on the periodic structure $V_{j(inup)}$):

$$\underline{V}_j^s = -\underline{W}\underline{U}^{-1}\underline{M}(\underline{I} + \underline{\Gamma})\underline{V}_j^i \tag{66}$$

where $V_j^i \equiv V_j$ (*inup*), matrix *I* denotes the identity matrix, $\Gamma$—the reflection matrix from the periodic surface, *M*—the matrix of conjugate coefficients of the distribution of base functions into Floquet harmonics, and the *W* and *U* matrices are determined by the Fourier transform of the Green function of the surrounding medium and the base functions used in the Galerkin method.

For the incident field $V_{ji}$ then we get:

$$\left[ \begin{array}{c} \underline{V}_j^{s+} + \underline{\Gamma}\underline{V}_j^i \\ \underline{V}_j^{s-} + (\underline{I} - \underline{\Gamma})\underline{V}_j^i \end{array} \right] = \left[ \begin{array}{cc} \underline{S}_{11}^{(j)} & \underline{S}_{12}^{(j)} \\ \underline{S}_{21}^{(j)} & \underline{S}_{22}^{(j)} \end{array} \right] \left[ \begin{array}{c} \underline{V}_j^i \\ 0 \end{array} \right] \tag{67}$$

What from the asymmetry of the reflection coefficient on the dielectric boundary and the continuity of the field tangent components on the boundary of two dielectrics gives the following matrix form $S^{(j)}$:

$$\left[ \underline{S}^{(j)} \right] = \left[ \begin{array}{cc} \underline{S}_{11}^{(j)} & \underline{S}_{12}^{(j)} \\ \underline{S}_{21}^{(j)} & \underline{S}_{22}^{(j)} \end{array} \right] = \left[ \begin{array}{cc} +\underline{\Gamma} - \underline{WU}^{-1}\underline{M}(\underline{I}+\underline{\Gamma}) & \underline{I} - \underline{\Gamma} - \underline{WU}^{-1}\underline{M}(\underline{I}-\underline{\Gamma}) \\ \underline{I} + \underline{\Gamma} - \underline{WU}^{-1}\underline{M}(\underline{I}+\underline{\Gamma}) & -\underline{\Gamma} - \underline{WU}^{-1}\underline{M}(\underline{I}-\underline{\Gamma}) \end{array} \right] \quad (68)$$

*6.1. Dielectric Limit*

The scattering matrix of the dielectric boundary meets the conditions of anti-reciprocity:

$$\underline{S}_{11}^{(j)} = -\underline{S}_{22}^{(j)} = \underline{R}^{(j)} \quad (69)$$

and it is symmetrical

$$\underline{S}_{12}^{(j)} = +\underline{S}_{21}^{(j)} = \underline{T}^{(j)} \quad (70)$$

and the individual sub-matrices of the scattering matrices correspond to the reflection matrices $R^{(j)}$ and transmission $T^{(j)}$.

$$\left[ \underline{S}^{(j)} \right] = \left[ \begin{array}{cc} \underline{R}^{(j)} & \underline{T}^{(j)} \\ \underline{T}^{(j)} & -\underline{R}^{(j)} \end{array} \right] . \quad (71)$$

The $R^{(j)}$ and $T^{(j)}$ reflection matrices have zero elements outside the diagonals, which means that there is no coupling between the different field harmonics (in the isotropic case considered here). The diagonal elements of these matrices are expressed by the mode impedances $\eta_k^{(up)}$ and $\eta_k^{(low)}$ of the harmonics $\Psi_k$ propagating in the upper and lower centers, respectively:

$$R_{kk} = \frac{\eta_k^{(low)} - \eta_k^{(up)}}{\eta_k^{(low)} + \eta_k^{(up)}} \quad (72)$$

$$T_{kk} = \frac{2\left(\eta_k^{(low)}\eta_k^{(up)}\right)^{1/2}}{\eta_k^{(low)} + \eta_k^{(up)}}. \quad (73)$$

*6.2. Dielectric Layer*

The remaining element of the structure (the dielectric layer) corresponds to the propagation of individual harmonics in the dielectric medium (let's assume $b_j$) without coupling between the harmonics. The scattering matrix of such a (homogeneous, isotropic) dielectric layer also has only non-zero elements on the diagonal and they are expressed by the appropriate field phase change for propagating harmonics or a field amplitude change for disappearing harmonics:

$$S_{kl}^{(j)} = \delta_{kl} exp(ik_{zkl}b_j) \quad (74)$$

## 7. Cascade Assembly of the Individual Elements of the Composite Antenna

Knowing the scattering matrices $S^{(j)}$ of individual elements of the system, the global matrix $S$ is determined according to the scheme presented in point 2. So for each element of the antenna structure, we determine the transmission matrix corresponding to this element.

$$\left[ \underline{T}^{(j)} \right] = \left[ \begin{array}{cc} \underline{T}_{11}^{(j)} & \underline{T}_{12}^{(j)} \\ \underline{T}_{21}^{(j)} & \underline{T}_{22}^{(j)} \end{array} \right] = \left[ \begin{array}{cc} \underline{S}_{21}^{(j)} - \underline{S}_{22}^{(j)}\underline{S}_{12}^{(j)-1}\underline{S}_{11}^{(j)} & \underline{S}_{22}^{(j)}\underline{S}_{12}^{(j)-1} \\ -\underline{S}_{21}^{(j)-1}\underline{S}_{11}^{(j)} & \underline{S}_{12}^{(j)-1} \end{array} \right] \quad (75)$$

Then we determine the global transmission matrix by the product of successive partial transmission matrices:

$$\underline{T} = \underline{T}^{(J)}\underline{T}^{(J-1)} \dots \underline{T}^{(2)}\underline{T}^{(1)} \quad (76)$$

and then we calculate the global scattering matrix according to the formula:

$$[\underline{S}] = \begin{bmatrix} \underline{S}_{11} & \underline{S}_{12} \\ \underline{S}_{21} & \underline{S}_{22} \end{bmatrix} = \begin{bmatrix} -\underline{T}_{22}^{-1}\underline{T}_{21} & \underline{T}_{22}^{-1} \\ \underline{T}_{11} - \underline{T}_{12}\underline{T}_{22}^{-1}\underline{T}_{21} & \underline{T}_{12}\underline{T}_{22}^{-1} \end{bmatrix}. \tag{77}$$

## 8. Conclusions

According to the procedure given above, it is practically necessary to determine the partial and global transmission matrices for the determination of the scattering matrices. In the case of specific antenna structures, various modifications of the above analysis are applied in order to obtain faster convergence of the applied numerical procedure and limitations on the memory capacity requirements used by the numeric program. Extremely important from the point of view of the effectiveness and accuracy of the procedure is the selection of the number of truncation $K$ of harmonics included in the numerical calculations and their order in the series of the field distribution. Usually, the order of harmonics is determined by the increase in the value of their propagation constants. The value of the number $K$ depends on the characteristic transverse and longitudinal distances that are normalized to the refractive index of the medium. The rule is that the smaller the distances between the individual cells of the periodic plane, the higher the harmonics should be considered in the field distribution, including also the decaying modes, especially important at small distances between individual planar antenna elements. The proposed solution enables the analysis of antenna arrays located on a multilayer dielectric. The presented procedure for the solution of electromagnetic field scattering on the antenna array is the original solution. Based on the derived dependencies, a computer program will be developed to analyze the parameters and characteristics of this type of antenna.

**Funding:** This research was funded by Military University of Technology, Faculty of Electronics, grant number 22-738.

**Informed Consent Statement:** Not applicable.

**Data Availability Statement:** Not applicable.

**Conflicts of Interest:** The author declares no conflict of interest.

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
