# Peer review of "Multilayer Dielectric Periodic Antenna Structure in a Cascade View"

_applsci, doi:10.3390/app12094185_

Round 1

Reviewer 1 Report

The article is very interesting. It contains a well-described mathematical apparatus. The purpose was clearly presented. The intentions of the article have been fulfilled. As noted by Author, very important from the point of view of efficiency and accuracy of the procedure is the choice of the number of K harmonic truncations considered in the numerical calculation and their order in the series of field distribution. The methodology defined by the Author can be applied to antenna theory, as presented in this paper based on multilayer periodic systems. The article is certainly worthy of publication (a visible contribution to applied science and to antenna theory).

In the reviewer's opinion, further consideration could be given:

1). Supplement the abstract with the purpose of the article.

2). Identify future research directions and application examples in the conclusion.

In terms of editing (proofreader's job):

1). In the keywords, the semicolon is in a different font color, otherwise redundant characters: ' '

2). Adjust (i.e., standardize) the font size – the reviewer think that, for example, the font on page 5, 6 are different sizes.

3). Adapt the style of the bibliography to the ‘Applied Sciences’ journal

4). Some words, e.g. floquet, are capitalized and lowercase (it would be good to standardize this).

5). Titles of some sections end with a period and others do not.

6). Align uneven spacing in text such as, inter alia, line 216.

The above-mentioned editorial shortcomings do not, in my opinion, affect the substantive value of the article, which I assess at a high level.

Decision: The article as is (after proofreader's correction) should be published.

Author Response

Dear Recensent, 

please find the the text of the articule attached. it covers all the changes so following both Your and other recensents remarks. All of them have been done with the option of tracking changes for your convienience. 

as a comment to all the recensents remarks please find responses below:

  1. Future directions and applications will be possible to determine when the computer program to calculate the results shall be finalised. This will also enable to include simulations. This is planned as a next step of the research. That will enable to use the proposed solution in practise to project and test multilayer antennas in dielectric layer.
  2. Advantage and comparison to previous models – there were no such models in the past (to the authors knowledge) therefore this is the very first approach to test the antenna arrays on multilayer dielectric.
  3. © in the matrices is used in order to determine and mark left hand side and right hand side of the figures.
  4. Introduction has been amended with additional information referencing contribution of the paper.
  5. Organisation of the paper has not been adjusted in the introduction because in the introduction whole scope of the paper has been covered.
  6. Order in the references - the article refers to different papers and books in the text sometimes more than once.
  7. Mathematical expression for A_mn ourier series

Fourier series

Distribution coeffcients

Reviewer 2 Report

The paper is in general written alright. However, the paper lacks in motivation, results section, and conclusion. My comments are as under.

  1. Line 32 and 46, why have you mentioned the article as a chapter?
  2. Enumerate the main contribution of the paper in the introduction section.
  3. Include organization of the paper at the end of introduction section.
  4. References are not cited in an ascending order. Please follow the instruction to authors’ on the Journal website.
  5. Line 90-92 need to be rephrased.
  6. Cite references for equation (7)-(8)
  7. A typo needs correction in equation (9).
  8. Write the mathematical expression for A_mn in (14).
  9. Consistency is lacking in the paper. e.g., italic x and non-italic x both are used to mean the same value. Make the symbols consistent.
  10. Correct ‘conditio’ in line 227
  11. What is meant by Chapter 1 in line 371.
  12. The paper lacks a check on the derived expression. Please include necessary simulation results for the derived expressions.
  13. Copy pasting shall be avoided. Please correct Reference 1 which is citing [32]

Author Response

(The authors gave the same response as above.)

Reviewer 3 Report

Please find my comments in the attached file.

Author Response

(The authors gave the same response as above.)

Round 2

Reviewer 2 Report

It would have been better to include simulation results in this paper itself. Nevertheless, I give it a pass.

Reviewer 3 Report

The implemented changes made the manuscript suitable for publication in the Applied Sciences Journal.